# GIS Models for Sustainable Urban Mobility Planning: Current Use, Future Needs and Potentials

**Xu Liu** [1,*] **, Peerawat Payakkamas** [1]**, Marc Dijk** [1] **and Joop de Kraker** [1,2]

1    Maastricht Sustainability Institute, Maastricht University, 6200 MD Maastricht, The Netherlands
2    Department of Environmental Sciences, Open Universiteit, 6419 AT Heerlen, The Netherlands
*    Correspondence: xu.liu@maastrichtuniversity.nl

**Abstract:** GIS models are currently available for a broad range of applications in mobility planning. However, it is not known how widespread the current use of GIS models is among European urban mobility planners, nor what their user experiences and needs are. There is therefore a risk that the development of GIS models for urban mobility planning will be mainly driven by technical possibilities and data availability rather than by the needs of the prospective users. To inform model developers and ensure a good match between model options and user needs, we conducted a survey investigating the current application of GIS models in urban mobility planning practice in Europe as well as model data availability and the needs and priorities of European mobility planners regarding GIS models. We received 51 valid responses from the transport departments of 42 cities from 21 European countries. For developers of GIS-based traffic models, the findings indicate that in Europe there is scope for wider adoption and further improvement. The models currently used are considered useful to support urban mobility planning, but more than 60% of the surveyed cities do not yet use them. Increased user-friendliness, in particular for non-experts, appears important to promote wider adoption. Availability of non-traditional types of data, such as real-time data or data at neighborhood level, is still limited in most cities, but this may rapidly change. Finally, there is also considerable interest in traffic models that integrate social and environmental aspects.

**Keywords:** GIS; traffic model; transport model; urban mobility planning; mobility policy; Europe

## 1. Introduction

In 2019, the European Commission launched the European Green Deal, which consists of a series of policies targeted to reach a climate-neutral Europe in 2050 [1]. For the transport sector, the specific objectives are to increase the uptake of zero-emission vehicles and to make sustainable alternative solutions available, while supporting digitalization and automation, and improving connectivity and access [2]. For cities, these objectives have been elaborated in the New European Urban Mobility Framework [3]. This policy framework mentions the importance of modeling "to support mobility decision-making in an integrated matter". It also emphasizes the use of urban mobility data to support sustainable urban mobility planning. The framework proposes that in this context not only typical mobility-related aspects such as road safety and congestion should be covered, but also environmental aspects, such as the emission of greenhouse gases and air pollution, as well as social aspects, such as access to mobility services and affordability of public transport [3]. Various studies showed that data-driven decision- and policy-making can help to improve the effectiveness of plans and policies [4–6]. Models play a major role in translating data into valuable information for decision- and policy-making [7,8]. For urban mobility planners, GIS models in particular can be an important help in achieving policy goals.

Due to their ability to process different types of data, graphic user interface and extensive map-based visualization options, GIS models are functional, cost-efficient and

user-friendly tools for (urban) mobility planning [9]. The currently available GIS models for mobility planning cover three main subjects: travel safety assessment, public transport management, and route planning [10]. For travel safety assessment, GIS models are integrated with decision support systems to simulate different traffic scenarios to predict the potential accidents and risks (e.g., Rahman et al. [11] and Rodrigues et al. [12]). For public transport management and planning, GIS models are, for example, used to analyze investment plans for public transport to determine how accessibility can be increased [13]. Route planning covers traveling route planning, public transport network planning, and safe walking and cycling route planning. GIS models are often adopted in this domain due to their capacity to integrate the processing of spatial data with network analysis [9].

The functionalities and possible applications of GIS models have rapidly evolved and are still growing. This also applies to the domain of urban mobility planning. An emerging application concerns GIS-based analyses that merge different types of mobility data (GPS data, mobile phone data, and location-based social media data) with spatial data and road networks. For instance, Droj et al. [10] utilized real-time traffic data in a network analysis to optimize public transport services and reduce traffic congestion in Oradea, Romania. Another emerging application concerns the integration of GIS models with other types of models. For example, Deng et al. [14] integrated Building Information Modeling (BIM) and GIS to assess traffic noise in outdoor and indoor environments, as determined by exterior and interior walls, in order to evaluate traffic regulations.

Hence, academic studies show that GIS models are currently available for a broad range of applications in mobility planning. However, it is not known what the needs and priorities of urban mobility planners are in this respect [15]. In fact, it is not even known how widespread the current use of GIS models among European urban mobility planners is, nor what their user experiences are. This means that there is a risk that the development of GIS models for urban mobility planning will be mainly driven by technical possibilities and potential data availability rather than by the needs of the prospective users and actual data availability. Thus, in order to guide future model development and to ensure a good match between model options and user needs, we conducted an exploratory survey to investigate the current use of GIS models in urban mobility planning practice in Europe, as well as the needs and future potentials. More specifically we addressed the following research questions: (1) How widespread is the current use of GIS models among European urban mobility planners? (2) What are their user experiences with these models? (3) What are their needs and priorities regarding GIS models? (4) What is the current availability of data for use in GIS models?

The paper is structured as follows. Section 2 describes our survey method, explaining how we collected and analyzed the data, with the results presented in Section 3. Finally, Section 4 discusses the main findings and limitations of our study, the implications for research and GIS model development, and Section 5 presents the conclusion.

## 2. Methods

To answer our research questions, we conducted an online survey. Online surveys have been proven to be as valid as paper-based surveys, easily combine different types of questions, are conveniently completed by participants, can be very widely distributed with little effort, and, lastly, help to avoid social desirability bias [16–18]. The questionnaire consisted of nineteen questions divided over four sections (Appendix B). Respondents needed 6–15 min to fill out the questionnaire. The first section covered basic information, including the respondent's working location and position with corresponding activities, based on closed-ended questions. The second section addressed research questions (1) and (2), and asked questions about the current use of GIS models in urban mobility planning practice, including motivations for model use and the type of data used. This section combined closed-ended and open-ended questions. Open-ended questions were employed to identify the varied concerns regarding GIS models based on each respondent's situation and experience. The third section addressed research question (4), and only targeted

respondents who work with data, such as data collection, analysis and modeling. The respondents were asked to indicate the availability, frequency, and reliability for nine different data types (Table 1), to examine the potential of traffic models based on new data combinations. As proposed by the New Urban Mobility Framework [3], we included the social and environmental dimension in addition to the mobility dimension. The selection of the data types representing these three dimensions was based on the literature [19–23]. The scales used in the questions were based on Tafidis et al. [24]. Finally, the fourth section addressed research question (3), and the respondents were asked to rate the relative importance of five factors or aspects that may be included in a GIS model: accessibility, livability, air quality, vehicle energy transition, and investment cost. The aim was to understand the priorities of European urban mobility planners and to provide guidance for developers of GIS-based traffic models.

**Table 1.** Nine types of mobility data relevant for use in GIS models.

| Dimensions | Types of Data | Sources |
|---|---|---|
| Mobility | Real-time traffic data<br>Public transport network coverage<br>Mobility networks<br>Travel distance to key services | [19–23] |
| Social | Traffic fatalities and injuries<br>Commuting travel time<br>Affordability of public transport | |
| Environmental | $PM_{2.5}$ air pollution at neighborhood level [1]<br>Greenhouse gas emissions at neighborhood level | |

[1] $PM_{2.5}$: fine inhalable particles, with diameter of maximum 2.5 μm.

Our study was exploratory in nature, and aimed at identifying broad patterns with respect to GIS model use in European urban mobility planning, without the aim to identify factors explaining the observed patterns. Therefore, for most of the responses, we use simple descriptive statistics to present the results, such as frequencies, means and standard deviations. We always indicate whether the results refer to individual respondents or cities, while specifying the total number involved ($N$). To determine the perceived relative importance of the five factors (accessibility, livability, air quality, vehicle energy transition, and investment cost), we employed analytic hierarchy process (AHP) [25]. This procedure allows to calculating the relative weight (importance) of multiple factors from pairwise comparison.

We conducted the online survey between September 2021 and February 2022 using the *Qualtrics* software platform (https://www.qualtrics.com/, accessed on 1 July 2021). Invitation letters ($N = 606$) with a hyperlink to the online questionnaire were distributed through email to the transport (or mobility) departments of European cities (or urban regions) and 56 responses were received. The mail addresses for the invitation letters were obtained from the participants list of the European Mobility Week 2021. Details about the responses and respondents are provided in Appendix A. We excluded five responses (marked in red in Appendix A) since the time taken to fill out the questionnaire was less than 5 min and therefore the response was considered not valid for the analysis. The final data set consisted of 51 valid responses, covering 42 cities from 21 European countries. The average number of residents in these cities is about 415,000, while most of the responding cities (75%) have a population size between 100,000 and 1,000,000 inhabitants. The number of responses is higher than the number of cities because sometimes more than one staff member of a city's transport department filled out the questionnaire. Figure 1 shows the geographic locations of the 42 cities, which cover most of Europe.

Out of the 51 respondents, 37% (19) are advisors, 22% (11) are data analysts, 22% (11) are policymakers, and 16% (8) are program managers, while only about 6% (3) are researchers. Thirteen respondents (25%) specified their job as mobility planner (3), depart-

ment officer (4), transport engineer (3), technologist (2) or GIS specialist (1). 13 respondents chose more than one position to specify their job. Activities of respondents cover the entire cycle from data collection, data analysis and model development, information support to policymakers, and traffic plan development, to monitoring and evaluation of policy measures (see Appendix A). This shows that urban mobility planning practice consists of a range of interconnected positions and activities associated with the development and evaluation of mobility plans, policies and measures. This being noted, we will use the terms urban mobility planners and planning throughout the paper to refer to the respondents and their activities.

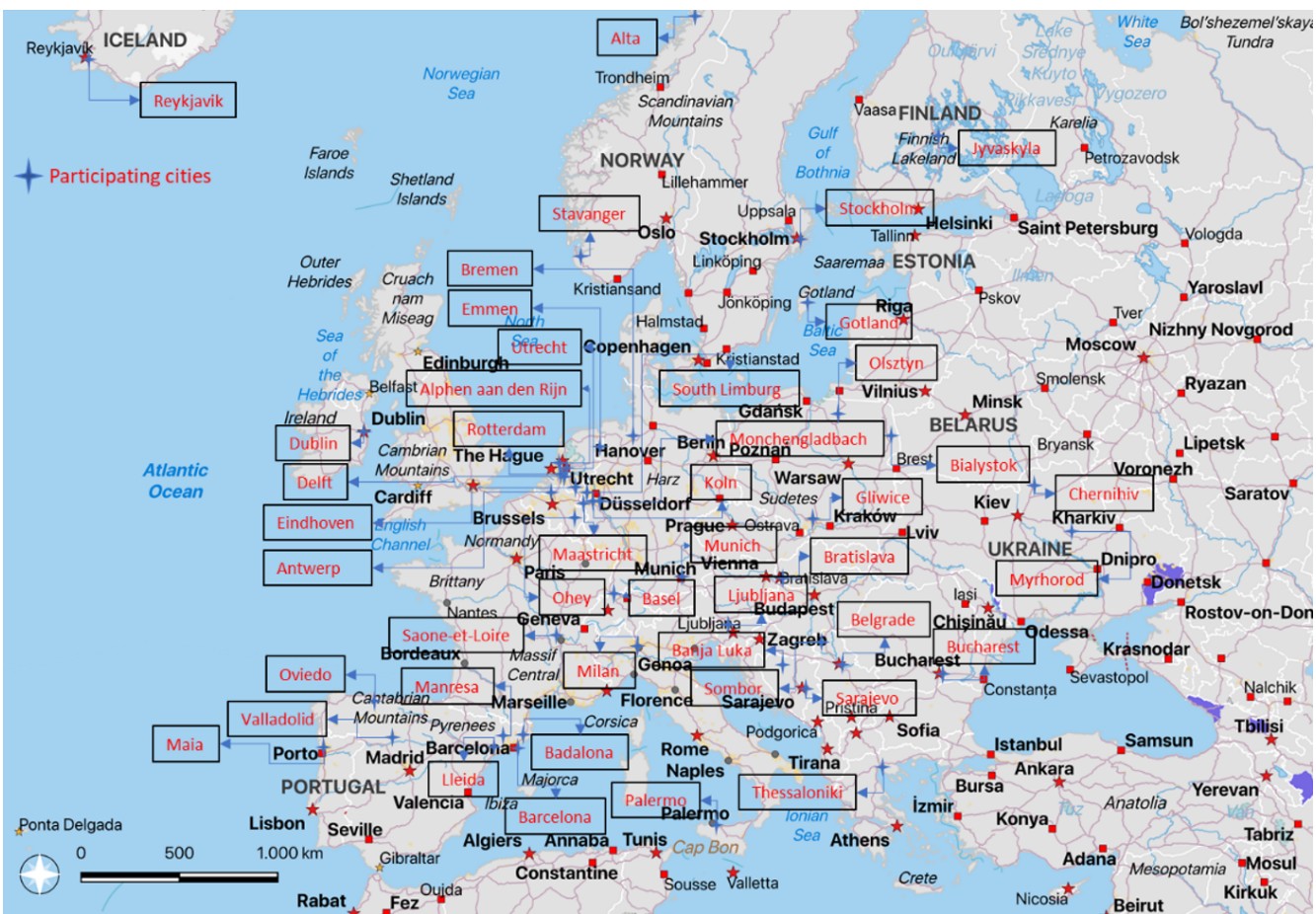

**Figure 1.** Location of the 42 cities or urban regions included in the survey (city name in square box).

## 3. Results

### 3.1. Current Use of GIS-Based Traffic Models in European Cities

According to the survey results, 37% (19) cities have experience with using traffic models for urban mobility planning and of these 84% (16) currently work with GIS-based traffic models. In 50% (8) of the cities, the use of these GIS models began already before 2010, the other half started using GIS models over the past 10 years (Figure 2). Most of the cities that work with GIS models are from Western and Southern European countries. In terms of data types used in the GIS models, GIS data, historical traffic data and survey data are most commonly used, whereas real-time traffic data and mobile phone data are still hardly used, while GPS data is not used by any of the cities in their GIS models.

There are several commercial GIS software tools that are used by more than one city. Basel, Badalona, and Barcelona employ TransCAD, a GIS-based tool for traffic analysis, transport modeling, and policy assessment. TransCAD is used by these cities for modeling of mid-term and long-term scenarios for mobility policy measures, assessing the impacts of mobility plans, and estimating air pollution. A mobility engineer from Badalona further specified that this model is used for logistics planning at a neighborhood level, as input for traffic regulation measures. Bucharest and Munich use PTV VISUM, a GIS tool to simulate traffic flows, for analyzing short-term policy measures and route networks, especially for regulating public transport. The biggest strength of PTV VISUM is that it provides a visual modeling interface and allows users to select and edit network objects in GIS maps [26]. Citylab's CUBE also provides an open GIS modeling mode for planners and engineers. A traffic data analyst from Milan specified that CUBE helps them to evaluate different policies at various time scales by simulating traffic flow in different scenarios.

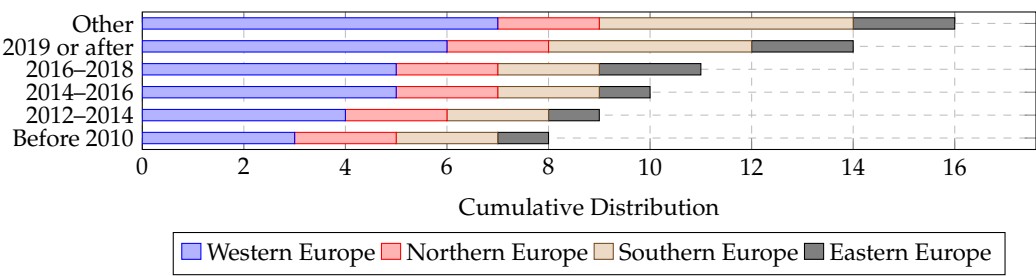

**Figure 2.** Starting year of GIS model use by region (*N* = 16 cities).

Table 2 shows the main reasons why cities started to use of traffic models. For the cities with more than one respondent, we chose the answers of the respondent who selected the most options. 'To get more information about traffic flow and trends for decision-making' and 'To use more actual data for evidence-based policy-making' are the two most frequently selected reasons, chosen by more than half of the cities. Ex-ante (prediction) and ex-post (evaluation) assessments of measures and policies was almost equally important as a motivation for GIS model use (selected by 7 and 8 cities respectively). Only one city started using a GIS model to examine its usefulness.

**Table 2.** Reasons to start using GIS models (more than one answer possible, *N* = 16 cities).

| Motivation | Number |
| --- | --- |
| To get more information about traffic flow and trends for decision-making | 11 |
| To use more actual data for evidence-based policy-making | 10 |
| To evaluate the implemented policies | 8 |
| To predict the impacts of policy measures | 7 |
| We were obliged to develop and use the model | 2 |
| To learn about/test the usefulness of such a model | 1 |
| It was offered to us for free by the government | 0 |
| Other (please specify) | 1 |

Concerning the perceived usefulness of GIS models in urban mobility planning, 11 out of 17 (65%) respondents from 16 cities (2 respondents are from the same city), considered that these models help them a lot or a great deal in urban mobility planning. 5 respondents (29%) regarded their models of moderate help, while only one respondent chose 'a little', explaining that their model is not suitable to assess accessibility.

### 3.2. Mobility Planners' Needs Concerning GIS-Based Traffic Models

All respondents, both GIS model users and non-users, were asked which type of information they considered GIS models should provide to support urban mobility planning. Table 3 shows that providing information about accessibility (78%), social aspects (64%),

and environmental aspects (60%) are the three main domains that GIS models should cover. Information on urban health, more specifically impacts on residents' health, is important for 42% of the respondents, whereas 22% of the respondents consider it important to integrate (the effects of) the energy transition into GIS-based analysis for mobility planning. 6 cities that all applied GIS models for mobility policy-making before, made use of the 'other' option to indicate more specific requirements. These ranged from integrating more factors to model trip planning behavior, such as costs of parking, scenery along the route and comfort level, to more detailed output data including driven speeds, traffic volume, and citizens' commuting habits for better planning of public roads and infrastructure transitions. Apart from these information requirements, nine respondents from different cities, of which 6 were from Eastern European countries, indicated the need for more financial support for model development.

**Table 3.** The type of information that respondents consider GIS models should provide to support urban mobility planning (more than one answer possible, *N* = 45 respondents).

| Options | Number |
| --- | --- |
| Provide information and insights about traffic flow and accessibility | 35 |
| Provide information about social aspects (e.g., residents opinions about new road constructions or transport poverty) | 29 |
| Provide information about environmental aspects (e.g., how does the new urban mobility policy or plan affect the local air quality) | 27 |
| Provide information about the impacts on residents' health | 19 |
| Integrate (the effects of) the energy transition into the model analysis | 10 |
| It is fine as it is | 1 |
| Other | 6 |

When asked about various aspects of user-friendliness of GIS models that require improvement, most of the respondents (76%) indicated that using the model should be made easier for staff who have less model and data processing knowledge. Providing more information at the neighborhood scale ranked second with 51%. Three other aspects of user-friendliness were less often chosen by the respondents: higher speed (36%), greater accuracy (27%), and more frequent upgrades (27%).

Following up on earlier questions about the need for information on social and environmental aspects, and information at the neighborhood scale, we asked the respondents to what extent and why they are (not) interested in having a GIS model that can evaluate the combined environmental and social effects of urban mobility policies and give results at a neighborhood level. Two-thirds of the respondents (68%) were 'extremely' or 'very interested' in this option. The main reason mentioned by the respondents was that this fits well with national and European environmental policies and could contribute to reaching greenhouse gas emission reduction goals. Another common reason mentioned (mostly by data analysts and advisors), is the need to provide more information to support plans and policies promoting a transition in travel behavior. About one-thirds (30%) of the respondents was only 'somewhat interested', for example, because they first want to have a better understanding of such a model or because they are satisfied with their current model.

### 3.3. Development Potential of GIS-Based Traffic Models

To explore potential development directions for GIS models, we asked respondents who mainly work with traffic data and models, about the availability, measurement frequency and reliability of nine types of data (Table 4 and Appendix C). And higher mean scores indicate better data availability, more frequent measurements, and a higher reliability of the data, whereas the size of the standard deviation (SD) indicates the degree of variation between the cities. The mean score for data availability ranged from 2.3 ('greenhouse gas emissions accounted at a neighborhood level') to 4.2 ('public transport network coverage'). Interestingly, both data types also have the lowest and highest mean scores, respectively,

for measurement frequency and reliability. In the case of data on 'greenhouse gas emissions accounted at neighborhood level', the low mean scores (3.2 for data frequency and 2.9 for data reliability) coincide with high standard deviations (1.4 for data frequency and 1.3 for data reliability), indicating large variation between cities, whereas the opposite is true for data on 'public transport network coverage'. The differences between these two data types appear to represent a broader pattern of higher data availability, measurement frequency and reliability for the more traditional and/or static types of traffic-related data, such as data on transport networks and traffic safety, and lower scores for these attributes for newer types of data with higher spatial or temporal resolution, such as environmental data at a neighborhood level and, especially for availability, real-time traffic data. These lower mean scores tend to coincide with relatively high standard deviations, indicating that availability, measurement frequency and reliability of these data are low for a major part of the responding cities, but high for a smaller group.

**Table 4.** Data availability, data measurement frequency, and data reliability for nine types of traffic-related data (mean and standard deviation, *N* = 23 respondents).

| Data Type | Data Availability [1.] | | Data Measurement Frequency [2.] | | Data Reliability [3.] | |
|---|---|---|---|---|---|---|
| | Mean | SD | Mean | SD | Mean | SD |
| Commuting travel time | 2.9 | 1.6 | 3.6 | 1.3 | 3.4 | 1.0 |
| Travel distance to key services | 3.3 | 1.5 | 3.2 | 1.3 | 3.5 | 0.9 |
| Affordability of public transport | 3.0 | 1.5 | 3.8 | 1.3 | 3.8 | 0.8 |
| Greenhouse gas emissions accounted at neighborhood level | 2.3 | 1.4 | 3.2 | 1.4 | 2.9 | 1.3 |
| $PM_{2.5}$ pollution accounted at neighborhood level | 2.9 | 1.6 | 3.2 | 1.5 | 3.2 | 1.4 |
| Mobility (road/cycle path/pedestrian path) networks | 3.7 | 1.3 | 3.9 | 1.2 | 3.8 | 0.9 |
| Public transport network coverage | 4.2 | 1.0 | 3.9 | 1.3 | 4.1 | 0.7 |
| Traffic fatalities and injuries | 3.8 | 1.2 | 3.6 | 1.3 | 3.9 | 0.9 |
| Real-time traffic data | 2.7 | 1.5 | 3.8 | 1.5 | 3.9 | 1.1 |

[1.] 1 = not available; 2 = available at a cost; 3 = available with special permission; 4 = freely available; 5 = freely available online [2.] 1 = measurements $\geq$ 10 years; 2 = 3–10 years; 3 = 1–3 years; 4 = annually; 5 = monthly/daily [3.] 1 = weak assumptions, significant inconsistency; 2 = debatable assumptions, considerable inconsistency; 3 = reasonable assumptions, moderate inconsistency; 4 = realistic assumptions, slight inconsistency; 5 = no assumptions, no inconsistency.

Finally, we sought a better insight in the perceived relative importance of factors in urban mobility planning and how this differs between cities. This could help to identify what should be covered by future, more integrated GIS-based traffic models. With an AHP analysis (see Section 2), relative importance scores were calculated for five factors: accessibility, livability, air quality, vehicle energy transition, and investment cost. Figure 3 shows the relative importance scores per city, as well as the mean value of each factor, based on 23 valid answers. Accessibility, air quality and livability rank highest among the respondents, whereas investment cost and the vehicle energy transition rank lowest. It appears that in most European cities, the traditional concerns of urban mobility planning (accessibility, air quality and livability) are still considered more important than the reduction of greenhouse gas emissions from the transport sector. However, perhaps more striking than this average rank order of the five factors, is the large variation between the responding cities. Each city seems to have its own, more or less unique, order of priorities in urban mobility planning.

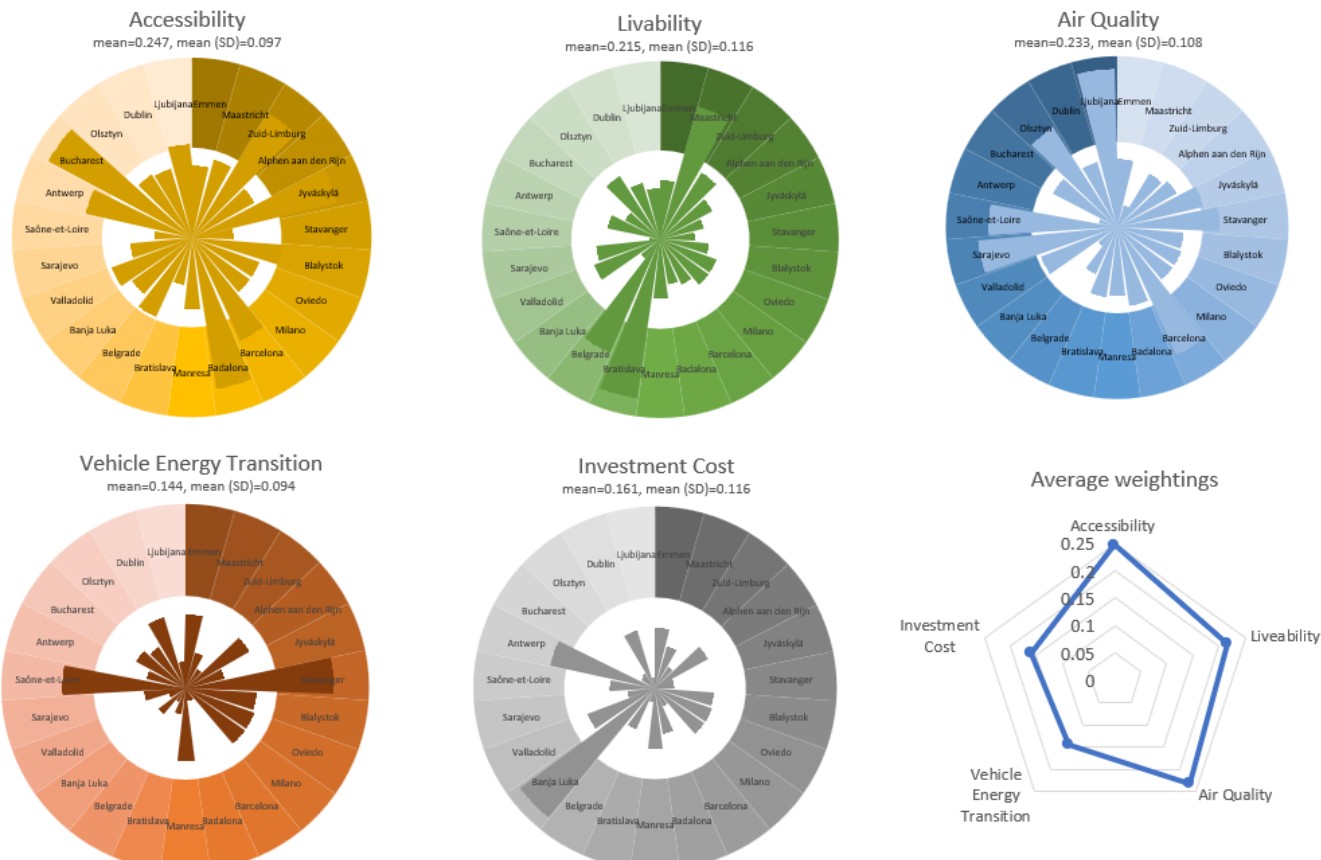

**Figure 3.** Perceived relative importance of different factors in urban mobility planning, per city and average score (*N* = 23 respondents).

## 4. Discussion

### 4.1. Major Findings and Implications for GIS Model Development

Our investigation made clear that currently the use of traffic models among European urban mobility planners is fairly widespread and established. These models are used in about 40% of the 42 European cities covered in our survey, mostly in Western and Southern Europe. This concerned predominantly (85%) GIS-based models, which were already in use for over 10 years in about half of the cities concerned. These models are used to support decision-making as well as policy development and evaluation, and the large majority of the respondents considered the models very useful for this purpose. Needs and interests of (prospective) users regarding the models, concern in the first place provision of information about accessibility, but there is also strong interest in information about social, environmental and health aspects of mobility, as well as information at neighborhood level. Current commercial traffic models generally center around management of traffic flows (e.g., TransCAD and VISUM). Fast, GIS-based models for calculating travel times, which could provide information about accessibility, have been developed, but have not been published yet or offered on the market [27].

Other needs and interests concerned improved user-friendliness, primarily (76%) in terms of easier use of the models by non-experts. Good (i.e., frequently measured and reliable) data for use in GIS models are available in most cities for more traditional data types, such as data on transport networks and traffic safety. However, for newer types of data, such as real-time traffic data and environmental data at neighborhood level, this is only the case for a minor part of the cities. In terms of main concerns, long-standing issues such as accessibility, air quality, and livability ranked highest, whereas the energy transition, a relatively new concern, ranked lowest. However, each city seems to have its own, more or less unique, order of priorities in urban mobility policy.

For developers of GIS-based traffic models, the findings indicate that in Europe there is scope for wider adoption and further improvement. The models currently used are considered useful to support urban mobility planning, but more than 60% of the surveyed cities do not yet use them. Increased user-friendliness, in particular for non-experts, appears important to promote wider adoption. There is also considerable interest in integrating more aspects (social and environmental) and types of data (neighborhood level). At the moment, the required data are well available in only a minor part of the cities, but this may change rapidly in the future given the strong emphasis of European Union (EU)'s policies on the integration of social (e.g., transport poverty) and environmental (e.g., decarbonization) aspects in sustainable urban mobility planning. This will also allow the use of GIS models that combine the more traditional and newer types of data. Given that cities differ considerably in terms of their priorities in urban mobility policy, the supporting GIS models should ideally cover a broad range of domains allowing the user to choose the options or modules that are locally of most interest.

### 4.2. Limitations and Research Needs

Our survey covered European cities quite well, both geographically and in terms of size. However, with 42 responding cities from 21 countries, it is still a limited sample. Given the low response rate, we therefore focused on broad patterns rather than details. Furthermore, to restrict the time required to complete the survey, we chose to include mostly closed questions. Consequently, we have a clear indication that the major part of European cities does not (yet) use traffic models to support urban mobility planning, but we do not yet know why this is the case, and what the city-specific reasons or barriers are. Likewise, it is now clear that there are considerable differences between cities in availability and quality of traffic-related data and in priorities in urban mobility planning, but we still lack deeper insight into the causes of these differences. These questions could be addressed with different a type of survey questionnaire designed to identify explanatory factors through advanced statistical analysis. Based on our current experiences, we expect, however, that the large sample size that would be required would be difficult to achieve. An alternative approach to acquire these insights would be to conduct in-depth case studies with a limited but highly diverse set of cities. This set could be selected from our sample of 42 European cities, of which the basic characteristics regarding GIS model use and needs are now known. These case studies could be combined with user participation in GIS model development, to ensure that further development of traffic models meets the needs and priorities of urban policymakers. User involvement in model development is strongly recommended, given the need to be adaptive to local requirements and conditions in the many decisions that are inherent to the development of models for decision and policy support. New developments in traffic models, such as high speed/high resolution models and models that use real-time data [27–30], come at a cost and depending on local budgets and priorities, this may be acceptable or not. Furthermore, the need and ways to deal with 'old' barriers to model use, such as the availability of certain types of data or data formats [31,32], or 'new' barriers, such as data privacy regulations [33], will differ between cities. Finally, in model development for decision and policy support there is always the dilemma between complex, integrated models covering multiple domains and producing high resolution output versus simple, user-friendly and easy-to-understand models [34–36]. Furthermore, in this case, involvement of prospective users is essential to make a choice.

### 5. Conclusions

We conducted this study to inform further development of GIS-based traffic models for urban mobility planning in Europe. As the study was the first of its kind, the survey responses from the transport departments of 42 cities from 21 European countries, provided novel insights into how widespread the current use of GIS models by urban mobility planners is, what the perceived usefulness of these models is, and what the availability and quality of traffic-related

model input data is. Furthermore, the study provided insight into the needs and priorities of mobility planners regarding GIS models, and how these vary between cities.

For developers of GIS-based traffic models, the findings indicate that in Europe there is scope for wider adoption and further improvement. The precise direction future model development should take is less clear, however, given the diversity in conditions, needs and priorities among European cities. We recommend that further in-depth research into this variation be done with a limited but diverse set of cities, possibly combined with GIS model development involving local, prospective users in Urban Living Lab type of settings.

**Author Contributions:** Conceptualization, X.L. and M.D.; methodology, X.L.; software, X.L.; validation, X.L. and M.D.; formal analysis, X.L. and P.P.; resources, X.L.; data curation, X.L. and P.P.; writing—original draft preparation, X.L. and P.P.; writing—review and editing, M.D. and J.d.K.; visualization, X.L. and P.P.; supervision, M.D. and J.d.K. All authors have read and agreed to the published version of the manuscript.

**Funding:** This research received no external funding.

**Institutional Review Board Statement:** Not applicable.

**Informed Consent Statement:** Not applicable.

**Data Availability Statement:** Detailed data will be available by requiring from the authors with permission of the respondents.

**Acknowledgments:** We acknowledge the support from all of our questionnaire respondents.

**Conflicts of Interest:** The authors declare no conflict of interest.

## Appendix A. Basic Information about the Responses and Respondents

The cities shown in red were excluded from the analysis.

**Table A1.** Basic information about the responses and respondents

| Number | City/Region | Country | Population | Position(s) | GIS Model Use (Year) | Response Date | Duration (Minutes) |
|---|---|---|---|---|---|---|---|
| 1 | Maastricht | The Netherlands | 121,565 | Advisor | No | 14 September 2021 | 27.23 |
| 2 | Maastricht | The Netherlands | 121,565 | Policymaker | Since before 2010 | 16 September 2021 | 16.33 |
| 3 | Maastricht | The Netherlands | 121,565 | Policymaker | Since 2012–2014 | 16 September 2021 | 10.13 |
| 4 | Maastricht | The Netherlands | 121,565 | Project manager | No | 17 September 2021 | 9.03 |
| 5 | Maastricht | The Netherlands | 121,565 | Policymaker; Advisor | No | 16 September 2021 | 16.55 |
| 6 | Emmen | The Netherlands | 107,113 | Advisor | Since 2012–2014 | 15 September 2021 | 12.23 |
| 7 | Emmen | The Netherlands | 107,113 | Advisor | No | 15 September 2021 | 6.40 |
| 8 | Rotterdam | The Netherlands | 651,157 | Advisor | No | 16 September 2021 | 6.32 |
| 9 | Rotterdam | The Netherlands | 651,157 | Researcher | No | 16 September 2021 | 4.77 |
| 10 | Zuid-Limburg | The Netherlands | 597,400 | Data analyst; Advisor; Researcher | No | 20 September 2021 | 7.85 |
| 11 | Eindhoven | The Netherlands | 231,642 | Policymaker | No | 21 September 2021 | 8.78 |
| 12 | Alphen aan de Rijn | The Netherlands | 110,986 | Data analyst | We use models since before 2010. Our actual model is since 2019. | 22 September 2021 | 16.07 |
| 13 | Utrecht | The Netherlands | 361,742 | Advisor; Designer | No | 27 September 2021 | 7.62 |

**Table A1.** *Cont.*

| Number | City/Region | Country | Population | Position(s) | GIS Model Use (Year) | Response Date | Duration (Minutes) |
|---|---|---|---|---|---|---|---|
| 14 | Delft | The Netherlands | 103,163 | GIS advisor/specialist | No | 19 October 2021 | 7.58 |
| 15 | Munich | Germany | 1,488,202 | Data analyst | Since 2014–2016 | 23 September 2021 | 12.42 |
| 16 | Bremen | Germany | 566,573 | Program manager | No | 24 September 2021 | 0.92 |
| 17 | Bremen | Germany | 566,573 | Program manager | No | 23 September 2021 | 17.72 |
| 18 | Cologne | Germany | 1,083,498 | Program manager | Other | 28 September 2021 | 2.08 |
| 19 | Mönchengladbach | Germany | 259,665 | Mobility planner | other (not sure) | 23 September 2021 | 6.93 |
| 20 | Oviedo | Spain | 214,883 | Advisor | Other | 30 September 2021 | 12.78 |
| 21 | Lleida | Spain | 137,856 | Policymaker; Program manager; Advisor; Researcher | No | 30 September 2021 | 8.38 |
| 22 | Barcelona | Spain | 1,620,343 | Program manager | 2020 | 1 October 2021 | 32.60 |
| 23 | Badalona | Spain | 217,741 | Data analyst; Mobility engineer | Since 2019 or after | 1 October 2021 | 20.08 |
| 24 | Manresa | Spain | 76,250 | Mobility Planner | Since before 2010 | 1 October 2021 | 7.97 |
| 25 | Valladolid | Spain | 299,715 | Policymaker | Since 2019 or after | 4 October 2021 | 30.83 |
| 26 | Palermo | Italy | 676,118 | Technical officer | No | 24 September 2021 | 8.75 |
| 27 | Milano | Italy | 1,399,860 | Policymaker; data analyst; Advisor | Since before 2010 | 30 September 2021 | 31.92 |
| 28 | Alta | Norway | 20,789 | Advisor | No | 24 September 2021 | 45.48 |
| 29 | Stavanger | Norway | 144,877 | Advisor | No | 28 September 2021 | 31.95 |
| 30 | Reykjavík | Iceland | 131,136 | Data analyst | Since before 2010 | 27 September 2021 | 8.77 |
| 31 | Reykjavík | Iceland | 131,136 | GIS manager | No | 28 September 2021 | 1.08 |
| 32 | Jyväskylä | Finland | 144,477 | Advisor; Project manager | No | 25 September 2021 | 7.62 |
| 33 | Gotland | Sweden | 58,595 | Mobility planner | No | 27 September 2021 | 146.78 |
| 34 | Stockholm | Sweden | 978,770 | Project coordinator | Yes (not sure which year) | 7 October 2021 | 1.78 |
| 35 | Lund | Sweden | 94,393 | Program manager | Since before 2010 | 20 October 2021 | 5.67 |
| 36 | Maia | Portugal | 135,306 | Municipal mobility technician | No | 27 September 2021 | 4.45 |

**Table A1.** *Cont.*

| Number | City/Region | Country | Population | Position(s) | GIS Model Use (Year) | Response Date | Duration (Minutes) |
|---|---|---|---|---|---|---|---|
| 37 | Białystok | Poland | 296,401 | Policymaker; Data analyst; Program manager | Passenger collecting system was fully implemented in 2003, E-ticketing was developed in years 2012–2014 | 29 September 2021 | 18.30 |
| 38 | Gliwice | Poland | 177,049 | Data analyst | No | 1 October 2021 | 86.98 |
| 39 | Olsztyn | Poland | 171,249 | City officer | No | 25 October 2021 | 19.97 |
| 40 | Myrhorod | Ukraine | 38,447 | Policymaker | No | 1 October 2021 | 10.63 |
| 41 | Chernihiv | Ukraine | 285,234 | Head of the Department of Transport, Transport Infrastructure and Communications | No | 1 October 2021 | 12.87 |
| 42 | Bratislava | Slovakia | 475,000 | Advisor | No | 1 October 2021 | 79.72 |
| 43 | Belgrade | Serbia | 1374,000 | Policymaker; Advisor; Program manager | No | 4 October 2021 | 6.37 |
| 44 | Sombor | Serbia | 47,623 | Data analyst; Program manager | No | 7 October 2021 | 0.88 |
| 45 | Banja Luka | Bosnia and Herzegovina | 138,963 | Policymaker; Program manager | No | 4 October 2021 | 14.25 |
| 46 | Sarajevo | Bosnia and Herzegovina | 275,524 | Program manager | No | 6 October 2021 | 11.98 |
| 47 | Saône-et-Loire | France | 551,493 | City officer | No | 7 October 2021 | 8.83 |
| 48 | Antwerp | Belgium | 523,248 | Data analyst | Since 2019 or after | 7 October 2021 | 31.58 |
| 49 | Ohey | Belgium | 5090 | Advisor | No | 7 October 2021 | 10.07 |
| 50 | Bucharest | Romania | 1,883,425 | Advisor; Engineering consultant | Since 2016–2018 | 8 October 2021 | 10.37 |
| 51 | Thessaloniki | Greece | 325,182 | Mobility officer | No | 11 October 2021 | 5.30 |
| 52 | Basel | Switzerland | 177,595 | Data analyst | Since before 2010 | 18 October 2021 | 22.65 |
| 53 | Dublin | Ireland | 554,554 | Mobility officer | No | 17 November 2021 | 6.23 |

**Table A1.** *Cont.*

| Number | City/Region | Country | Population | Position(s) | GIS Model Use (Year) | Response Date | Duration (Minutes) |
|---|---|---|---|---|---|---|---|
| 54 | Dublin | Ireland | 554,554 | Project manager; Project Engineer | No | 22 November 2021 | 13.32 |
| 55 | Ljubljana | Slovenia | 295,504 | Advisor | Since before 2010 | 1 February 2022 | 12.22 |
| 56 | European Commission | European Commission | N | Advisor; Data analyst | No | 13 October 2021 | 17.22 |

## Appendix B. Survey Questionnaire

Note: For each question, we indicated the number of respondents (*N*), and for each of the answer options, the number of respondents that chose this option (number in red). The numbers refer to all 56 responses that were received.

### *Appendix B.1. Introduction*

Computer models are regularly used in planning and policy-making for ex-ante and ex-post assessments. They can be helpful to evaluate different plans and policy options and assess the effects of policy measures. With respect to urban mobility planning, integrated 'Geographical Information System' (GIS) models could process location-based data (i.e., GIS data, GPS data), and integrate indicators such as accessibility, $CO_2$ emissions and aspects of health, and visualize the results for policymakers.

You have been selected to complete the following questionnaire which aims to understand what you think about integrated GIS models used in urban mobility planning and policy-making and what you expect from these models for your work.

Your contribution is highly appreciated. The questionnaire takes 8–12 min and there is no right or wrong answer. If there is a question that you prefer not to answer, please skip it and move on to the next. The research is scientific and has no profit-seeking purposes. Your data will be anonymized and treated confidentially.

### *Appendix B.2. Part 1: Basic Information*

Q1. At which city do you work? (*N* = 56)

Q2. Which label best indicates your position in the municipality? (multiple answers are possible) (*N* = 56)

- Policymaker (1) 11
- Data analyst (2) 13
- Program manager (3) 13
- Advisor (4) 19
- Researcher (5) 3
- Other (please specify) (6) 19

Q3. What does your work involve? (multiple answers are possible) (*N* = 55)

- Survey data collection (1) 17
- Traffic data collection (2) 23
- Survey data analysis (3) 18
- Real-time traffic data monitoring and analysis (4) 9
- Traffic plan development (5) 28
- Policy decision-making (6) 15
- Providing knowledge and information to policymakers (7) 34
- Communicating and cooperating with different work groups (8) 40

–   Monitoring and evaluating policy measures (9) 24
–   Model development (10) 19
–   Other (please specify) (11) 7

*Appendix B.3. Part 2: Current Use of GIS Models in Urban Mobility Planning and Policy-Making*

Q4.  Have you ever worked with a traffic model for urban mobility policy-making? (*N* = 56)

–   Yes (please simply describe the model) (1) 19
–   No (2) 37
–   Other (please specify) (3) 0

Q5.  Do you currently work with a GIS model for urban mobility policy-making?

–   Yes (1) 17
–   No (2) 35
–   Other (please specify) (3) 4

*Skip To: Q12 If 5. Do you currently work with a GIS model for urban mobility policy-making? = No*

Q6.  Please simply describe this model for and how it is applied in urban mobility policy-making: (*N* = 16)

Q7.  What was your motivation to start working with this model? (multiple answers are possible) (*N* = 16)

–   We wanted to use it to predict the impacts of policy measures for urban mobility plans and policy development (1) 7
–   We wanted to use it to evaluate the implemented policies and to see the impacts of these policy measures (2) 8
–   We wanted to get more information about traffic flow and trends based on data for decision-making (3) 11
–   We wanted to use more actual data for evidence-based urban mobility policy-making (4) 10
–   We wanted to learn about/test the usefulness of such a model (5) 1
–   We were obliged to develop and use the model (6) 2
–   It was offered to us for free by the national government (7) 0
–   It was offered to us for free by the provincial government (8) 0
–   It was offered to us through a consultancy as part of another project (9) 0
–   Other (please specify) (10) 1

Q8.  Since when has your municipality worked with this model? (*N* = 17)

–   Since before 2010 (1) 7
–   Since 2012–2014 (2) 2
–   Since 2014–2016 (3) 1
–   Since 2016–2018 (4) 1
–   Since 2019 or after (5) 3
–   Other (please specify) (6) 3

Q9.  What types of data do you use as inputs to this model? (multiple answers are possible) (*N* = 17)

–   GIS data (1) 14
–   GPS data (2) 0
–   Mobile phone data (3) 2
–   Real-time traffic data (4) 3
–   Historical traffic data (5) 12
–   Survey data (6) 10
–   Other (please specify) (7) 1

Q10. What are the sources of your data? (multiple answers are possible) (*N* = 17)

–    Central Bureau of Statistics (1) 5
–    Local road sensor cameras (2) 9
–    Public transport cards (3) 4
–    Social media (4) 0
–    Other companies (i.e., mobile phone data provided by *Vodafone*) (5) 4
–    Other (please specify) (6) 10

Q11. To what extent do you think this model contributes to urban mobility policy development? (*N* = 17)

–    A very great deal (1) 2
–    A lot (2) 9
–    A moderate amount (3) 5
–    A little (4) 1
–    Not at all (5) 0

Q12. GIS model users: Which aspect of GIS models do you think should be improved? Non-users: What knowledge would you want to get from these models? (multiple answers are possible) (*N* = 45)

–    Provide more information and insights about traffic flow and accessibility (1) 35
–    Provide more information about social aspects (e.g., residents opinions about new road constructions or transport poverty) (2) 29
–    Provide more information about environmental aspects (e.g., how does the new urban mobility policy or plan affect the local air quality) (3) 27
–    Provide more information about the impacts on residents' health (4) 19
–    Integrate (the effects of) the energy transition into the model analysis (5) 10
–    Provide more financial support for model development (6) 9
–    It is fine as it is (8) 2
–    Other (please specify) (7) 6

Q13. Which aspects of user-friendliness of GIS models do you think should be improved? (multiple answers are possible) (*N* = 45)

–    Make model use easier for staff who have less model and data processing knowledge (1) 34
–    The model should provide data and information at the neighborhood scale (2) 23
–    The model should process the data faster (3) 16
–    The accuracy of information given by the model should be improved (4) 12
–    The model should be upgraded more frequently (5) 12
–    It is fine as it is (7) 1
–    Other (please specify) (6) 4

Q14. To what extent you are interested in having a GIS model that can evaluate the combined environmental and social effects of urban mobility policies and give (visualized) results at neighborhood level? (*N* = 47)

–    Extremely interested (1) 10
–    Very interested (2) 22
–    Somewhat interested (3) 14
–    Not so interested (4) 0
–    Not at all interested (5) 1

Q15. Could you shortly explain why are you interested/not interested in having a GIS model that can evaluate the combined environmental and social effects of urban mobility policies and give (visualized) results at neighborhood level? (*N* = 29)

*Appendix B.4. Part 3: Data Availability, Measurement Frequency, and Reliability (N = 23)*

Q16. Data availability

| | Not available (1) | Available at a cost (2) | Available with special permission (3) | Freely available (4) | Freely online available (5) |
|---|---|---|---|---|---|
| Commuting travel time (1) | ○ | ○ | ○ | ○ | ○ |
| Travel distance to key services (2) | ○ | ○ | ○ | ○ | ○ |
| Affordability of public transport (3) | ○ | ○ | ○ | ○ | ○ |
| Greenhouse gas emissions accounted at neighborhood level (4) | ○ | ○ | ○ | ○ | ○ |
| $PM_{2.5}$ pollution accounted at neighborhood level (5) | ○ | ○ | ○ | ○ | ○ |
| Mobility (road/cycle path/pedestrian path) networks (6) | ○ | ○ | ○ | ○ | ○ |
| Public transport network coverage (7) | ○ | ○ | ○ | ○ | ○ |
| Traffic fatalities and injuries (8) | ○ | ○ | ○ | ○ | ○ |
| Real-time traffic data (9) | ○ | ○ | ○ | ○ | ○ |

Q17. Data measurement frequency

| | Measurements $\geq$ 10 years (1) | 3–10 years (2) | 1–3 years (3) | Annually (4) | Monthly/daily (5) |
|---|---|---|---|---|---|
| Commuting travel time (1) | ○ | ○ | ○ | ○ | ○ |
| Travel distance to key services (2) | ○ | ○ | ○ | ○ | ○ |
| Affordability of public transport (3) | ○ | ○ | ○ | ○ | ○ |
| Greenhouse gas emissions accounted at neighborhood level (4) | ○ | ○ | ○ | ○ | ○ |
| $PM_{2.5}$ pollution accounted at neighborhood level (5) | ○ | ○ | ○ | ○ | ○ |
| Mobility (road/cycle path/pedestrian path) networks (6) | ○ | ○ | ○ | ○ | ○ |
| Public transport network coverage (7) | ○ | ○ | ○ | ○ | ○ |
| Traffic fatalities and injuries (8) | ○ | ○ | ○ | ○ | ○ |
| Real-time traffic data (9) | ○ | ○ | ○ | ○ | ○ |

Q18. Data reliability (1 = weak assumptions/significant inconsistency; 2 = debatable assumptions/considerable inconsistency; 3 = reasonable assumptions/moderate inconsistency; 4 = realistic assumptions/slight inconsistency; 5 = no assumptions/no inconsistency)

| | 1 (1) | 2 (2) | 3 (3) | 4 (4) | 5 (5) |
|---|---|---|---|---|---|
| Commuting travel time (1) | ○ | ○ | ○ | ○ | ○ |
| Travel distance to key services (2) | ○ | ○ | ○ | ○ | ○ |
| Affordability of public transport (3) | ○ | ○ | ○ | ○ | ○ |
| Greenhouse gas emissions accounted at neighborhood level (4) | ○ | ○ | ○ | ○ | ○ |
| $PM_{2.5}$ pollution accounted at neighborhood level (5) | ○ | ○ | ○ | ○ | ○ |
| Mobility (road/cycle path/pedestrian path) networks (6) | ○ | ○ | ○ | ○ | ○ |
| Public transport network coverage (7) | ○ | ○ | ○ | ○ | ○ |
| Traffic fatalities and injuries (8) | ○ | ○ | ○ | ○ | ○ |
| Real-time traffic data (9) | ○ | ○ | ○ | ○ | ○ |

This part is only for those who are working with data (i.e., data collection, data analysis, model development). The aim is to assess the availability, measurement frequency and reliability of data that could be used to develop or apply a GIS model for urban mobility policy-making.

*Appendix B.5. Part 4: Comparative Importance of Different Factors (N = 40)*

In this part, we want to investigate the importance of different factors in urban mobility policy-making. This could to help to identify what should be covered by an integrated sustainability assessment GIS model for urban mobility policy-making.

Q19. Please compare the following factors:

If you compare for example accessibility and livability and you click the dot in the middle, it means that you consider both factors equally important. If you click the dot closer to accessibility, it means that you consider accessibility more important than livability. If you click the dot closer to livability, it means that you consider livability more important than accessibility.

| | 1 (1) | 2 (2) | 3 (3) | 4 (4) | 5 (5) | 6 (6) | 7 (7) | 8 (8) | 9 (9) | |
|---|---|---|---|---|---|---|---|---|---|---|
| Accessibility | ◯ | ◯ | ◯ | ◯ | ◯ | ◯ | ◯ | ◯ | ◯ | Livability |
| Livability | ◯ | ◯ | ◯ | ◯ | ◯ | ◯ | ◯ | ◯ | ◯ | Air quality |
| Air quality | ◯ | ◯ | ◯ | ◯ | ◯ | ◯ | ◯ | ◯ | ◯ | Accessibility |
| Vehicle energy transition | ◯ | ◯ | ◯ | ◯ | ◯ | ◯ | ◯ | ◯ | ◯ | Accessibility |
| Livability | ◯ | ◯ | ◯ | ◯ | ◯ | ◯ | ◯ | ◯ | ◯ | Vehicle energy transition |
| Vehicle energy transition | ◯ | ◯ | ◯ | ◯ | ◯ | ◯ | ◯ | ◯ | ◯ | Air quality |
| Accessibility | ◯ | ◯ | ◯ | ◯ | ◯ | ◯ | ◯ | ◯ | ◯ | Investment cost |
| Investment cost | ◯ | ◯ | ◯ | ◯ | ◯ | ◯ | ◯ | ◯ | ◯ | Livability |
| Air quality | ◯ | ◯ | ◯ | ◯ | ◯ | ◯ | ◯ | ◯ | ◯ | Investment cost |
| Investment cost | ◯ | ◯ | ◯ | ◯ | ◯ | ◯ | ◯ | ◯ | ◯ | Vehicle energy transition |

## Appendix C. Percentage of Respondents per Answer Option on Data Availability, Data Measurement Frequency, and Data Reliability for GIS Model Development and Operation

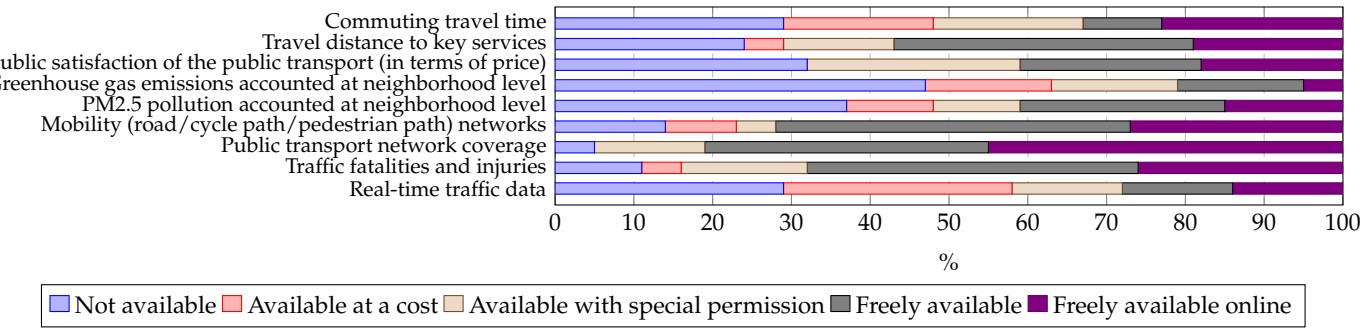

**Figure A1.** Data availability (*N* = 23).

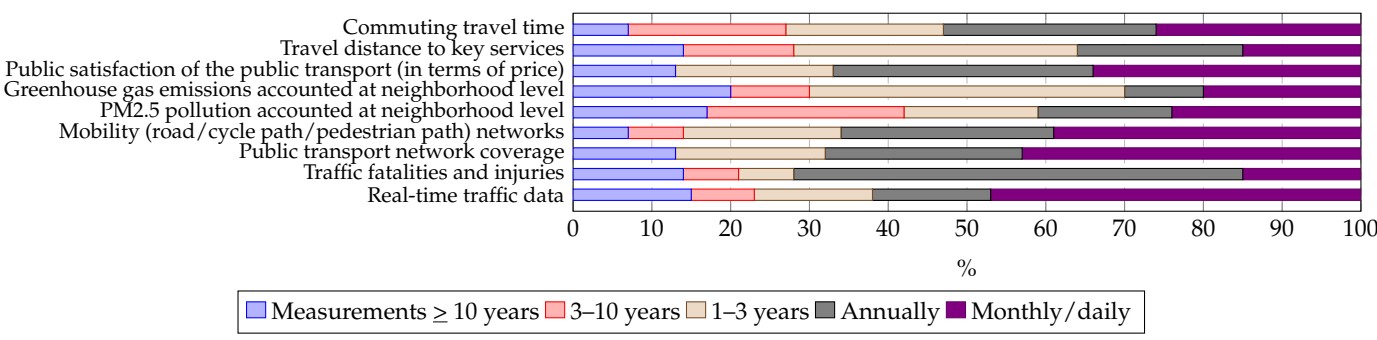

**Figure A2.** Data measurement frequency (*N* = 23).

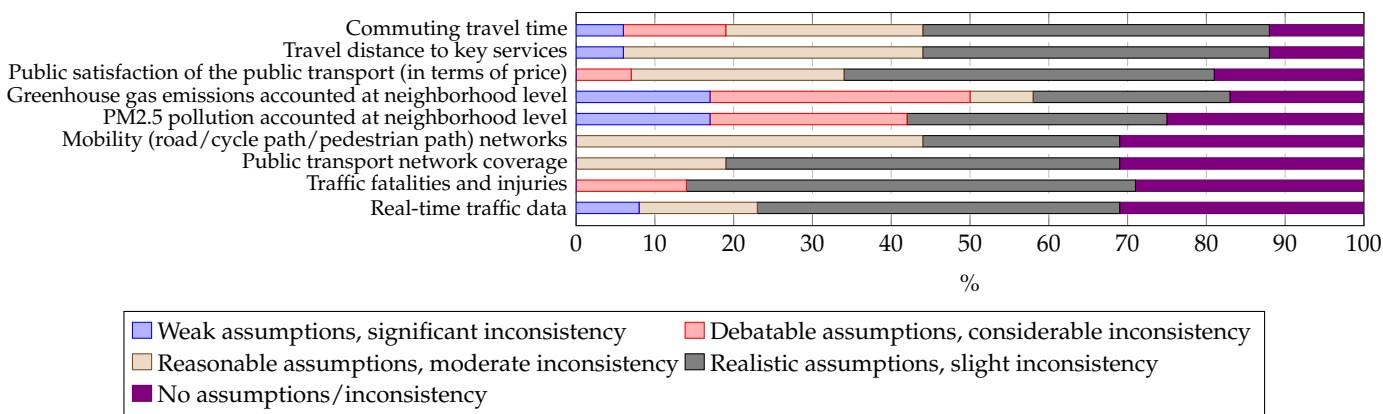

**Figure A3.** Data reliability (*N* = 23).

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
