# Peer review of "GIS Models for Sustainable Urban Mobility Planning: Current Use, Future Needs and Potentials"

_futuretransp, doi:10.3390/futuretransp3010023_

Round 1

Reviewer 1 Report

This paper analyzed surveys provided by transport departments of 42 cities from 21 European countries. It tried to reveal the current statuses on GIS models. For methodology part, it tried to use quantitative methods to describe qualitative data. Besides the efforts, I see this paper is more likely to be a report. Please see my comments below.

Comments 1:

Figure 2 is hard to read. CDF plots can be better.

Comments 2:

Figure 3 is too general. 17 respondents are less than 30. This is not representative in terms of statistics. Recommend deleting this.

Comments 3:

Exploring different factors is good, but in Figure 4, this sunburst graph is not suitable here.

Comments 4:

Very limited analysis was conducted. There can be more analysis. For example, cross-tabulation. Please refer to more policy-related papers. For example, Examining municipal guidelines for users of shared E-Scooters in the United States.

Author Response

Reviewer

Comments and Suggestions

Reply

1

1)     Figure 2 is hard to read. CDF plots can be better.

This suggestion has been taken by the authors. We changed figure to CDF plots.

2)     Figure 3 is too general. 17 respondents are less than 30. This is not representative in terms of statistics. Recommend deleting this.

This suggestion has been taken by the authors.

We deleted figure 3 in the paper.

3)     Exploring different factors is good, but in Figure 4, this sunburst graph is not suitable here.

As indicated in the text (Results and Discussion), the main purpose of the sunburst graph is to show the diversity among cities in how the respondents weighted the five factors, highlighting that modelers should consider the differences in local priorities when they develop GIS models for various cities.

4)     Very limited analysis was conducted. There can be more analysis. For example, cross-tabulation. Please refer to more policy-related papers. For example, Examining municipal guidelines for users of shared E-Scooters in the United States.

We did several different analyses including cross-tabulation and correlation analysis but no insightful results were found, so we did not present it in our results.

We added more policy-related papers as references in the paper:

-Ma, Q., Yang, H., Ma, Y., Yang, D., Hu, X., & Xie, K. (2021). Examining municipal guidelines for users of shared E-Scooters in the United States. Transportation research part D: transport and environment, 92, 102710.

- Sun, Z., Huang, T., & Zhang, P. (2020). Cooperative decision-making for mixed traffic: A ramp merging example. Transportation research part C: emerging technologies, 120, 102764.

- Zhu, Y., Wu, Q., & Xiao, N. (2022). Research on highway traffic flow prediction model and decision-making method. Scientific reports, 12(1), 19919.

Reviewer 2 Report

The study investigates the current application of GIS models in urban mobility planning practice including data availability and the needs of mobility planners. Overall, the paper is well-written and of interest to the audience of the journal. However, the paper has some issues that should be improved.

In the introduction, the authors describe the problem situation with some reference to previous research. Then, line 73 states: “To answer our research questions … “. The introduction lacks a presentation of the research questions and an outline of the theoretical background or/and the assumptions that underlies the study’s design and conduct. The paper states that “The aim was to understand the priorities of European urban mobility planners and to provide guidance for developers of GIS-based traffic models”, however, it remains unclear what this understanding concerns and how it is achieved. For this purpose, it could be advisable to develop a theory section that describes a background that help the authors to understand the results of the survey.

Such background might also be able to explain how the questionnaire has been developed. The method description lacks a motivation of the design of the study, how the questions were developed and how and why the scales for the answers has been selected. The authors might also reflect a bit on the issues of small n-studies and why statistical analysis might not be a good choice in such cases. Moreover, the analysis section should elaborate on how the collected data was analysed and interpreted. Since no quantitative analysis has been performed, the qualitative approach should be outlined in more detail.

In addition, the analysis of the collected data should be improved. A stronger analytical approach and theoretical background could be useful to analyse and interpret the survey data. Although the discussion section tries to overcome this issue, it misses to relate the findings to previous research in a convincing manner. The author could also consider to reflect on to what extend the findings from the urban context can relate to a more rural context in the participating countries.

Finally, as a matter of taste, I would suggest to present the conclusions as an individual section separated from the discussion.

Best regards!

Author Response

Reviewer

Comments and Suggestions

Reply

2

1)     In the introduction, the authors describe the problem situation with some reference to previous research. Then, line 73 states: “To answer our research questions … “. The introduction lacks a presentation of the research questions and an outline of the theoretical background or/and the assumptions that underlies the study’s design and conduct.

This suggestion has been taken by the authors. We now made our research questions explicit (lines 68–72).

2)     The paper states that “The aim was to understand the priorities of European urban mobility planners and to provide guidance for developers of GIS-based traffic models”, however, it remains unclear what this understanding concerns and how it is achieved. For this purpose, it could be advisable to develop a theory section that describes a background that help the authors to understand the results of the survey. Such background might also be able to explain how the questionnaire has been developed.

We elaborated and specified more about our research questions in the introduction (lines 68–72). We also elaborated more about the survey questionnaire and how it is designed to answer our research questions in the methods section (lines 78–102).

3)     The method description lacks a motivation of the design of the study, how the questions were developed and how and why the scales for the answers has been selected.

We now elaborate more about our research method, including the structure of the questionnaire, the design of different types of questions, and the scales (lines 78–102).

4)     The authors might also reflect a bit on the issues of small n-studies and why statistical analysis might not be a good choice in such cases.

 We added a clarification about the nature of our analysis, which is descriptive (identify broad patterns, e.g., in use and perceived usefulness of GIS traffic models), and in the analysis therefore limited to using descriptive statistics (lines 103–108). We also added a cautionary remark about the interpretation of the results, which, given the low N, should focus on broad patterns rather than details (lines 103–108).

5)     the analysis section should elaborate on how the collected data was analysed and interpreted. Since no quantitative analysis has been performed, the qualitative approach should be outlined in more detail.

We added a clarification about the nature of our analysis, which is descriptive (identify broad patterns, e.g., in use and perceived usefulness of GIS traffic models), and not explanatory (i.e., not aimed at identifying factors explaining the observed patterns) (lines 103–108).

6)     the analysis of the collected data should be improved. A stronger analytical approach and theoretical background could be useful to analyse and interpret the survey data. Although the discussion section tries to overcome this issue, it misses to relate the findings to previous research in a convincing manner.

In the introduction we state that knowledge is lacking about the spread of GIS model use and data availability, as no previous research on this topic has been conducted. We have now added a sentence to the conclusion stressing the ‘first-time’ nature of our study (lines 58–64).

As explained above, we therefore conducted an exploratory study to identify broad patterns. We have now elaborated the point of the need for more detailed and explanatory-oriented follow-up studies to the section on limitations and research needs (lines 286–315).

7)     The author could also consider to reflect on to what extend the findings from the urban context can relate to a more rural context in the participating countries.

Thanks for the suggestion. This might be an interesting area of future research, but it is beyond the explicitly urban scope of this study.

8)     Finally, as a matter of taste, I would suggest to present the conclusions as an individual section separated from the discussion.

This suggestion has been taken by the authors.

We present the conclusion now as an individual section.

Reviewer 3 Report

Title: GIS Models for Sustainable Urban Mobility Planning: Current Use, Future Needs and Potentials

Journal: Future Transp

This article surveys the use of GIS models in transportation planning of European cities by means of questionnaires, and makes statistical analysis based on the survey results, and puts forward the current situation and future development direction. The method used in this paper is relatively simple, and the descriptive statistical method is used. My concerns are listed as follows.

1.     The validity rate of the questionnaire is too low. A total of 606 questionnaires have been issued, but only 15 of them are valid. Besides, the number of samples is insufficient.

2.     In Lines 92, 93, and 94, there are only 51 questionnaires, but the sum of the numbers is 52.

3.     The reasons for the selection of indicators in Table 1 are not explained clearly.

4.     In Section 3.1, the number of cities studied is N=16. However, in Figure 3, the object of study is institutions, making N=17.

5.     In Line 167, what does ‘six cities, all GIS model users’ mean?

6.     In Line 202, ‘the low mean scores coincide with relatively high standard deviations’ is too subjective.

7.     Line 229. The data obtained from the survey cannot support the conclusion that GIS models are generally used.

8.     My other comments focus on the presentation.

(a) Line 20. The words “climate neutral” should be changed into “climate-neutral”.

(b) Line 29. such as the emission of greenhouse gases and air pollution

(c) Line 31. policy-making

(d) Line 43. how accessibility

(e) Line 50. and location-based social media data

(f) Line 51. real-time

(g) Line 107. the current use

(h) Line 122. calculating

(i) Lines 139, 184, 199, 209, 238, and 243. at a neighborhood level

(j) Line 172. six were from

(k) Line 184. Two-thirds 

(l) Line 189. One-thirds

(m) Line 190. For example,

(n) Line 196. And higher

(o) Line 201. In the case of 

(p) Line 203. large variation 

(q) Line 211. are low 

(r) Line 214. could help 

(s) Line 234. decision-making 

(t) Lines 239 and 251. user-friendliness

(u) Line 260. choose the 

Author Response

Reviewer

Comments and Suggestions

Reply

3

1)     The validity rate of the questionnaire is too low. A total of 606 questionnaires have been issued, but only 15 of them are valid. Besides, the number of samples is insufficient.

The valid questionnaire number is 51, not 15. Although a response rate of 8% (51 out of 606) is low, this tells us more about the usefulness of the mailing list we used than about the validity of the responses. With 51 responses from 42 cities and 21 European counties, and a balanced coverage in terms of city size and geographic location, the responses can be considered useful to answer our research questions in this exploratory study.

2)     In Lines 92, 93, and 94, there are only 51 questionnaires, but the sum of the numbers is 52.

This is based on a multiple-option question, which means that respondents could choose more than one role. For instance, a respondent could choose the options of advisor, data analyst, and mobility planner. This is why the results sum up for this question as 52+13=65.

3)     The reasons for the selection of indicators in Table 1 are not explained clearly.

We added a short explanation in the text about the basis of our selection, including the references (lines 92–96).

4)     In Section 3.1, the number of cities studied is N=16. However, in Figure 3, the object of study is institutions, making N=17.

The number of cities is indeed 16, but 2 respondents from the same city both validly answered this question (we now clarified this in the paper, see lines 171–175).

5)     In Line 167, what does ‘six cities, all GIS model users’ mean?

We edited the sentence to make it clearer. The sentence now is: ‘Six cities that all applied GIS models for mobility policy making before, made use of the ‘other’ option to indicate more specific requirements.’ (lines 183–185)

6)     In line 202, ‘the low mean scores coincide with relatively high standard deviations’ is too subjective.

To avoid unnecessary repetition, we prefer not to spell out what the reader can see in table 4:

-        Availability, range of SD is 1.0-1.6, SD of GHG is 1.4

-        Frequency, range of SD is 1.2-1.5, SD of GHG is 1.4

-        Reliability, range of SD is 0.7-1.4, SD of GHG is 1.3

7)     Line 229. The data obtained from the survey cannot support the conclusion that GIS models are generally used.

We conclude that “that currently the use of traffic models among European urban mobility planners is fairly widespread and established”. This conclusion is well supported by our findings  that in 40% of the 42 European cities in the survey traffic models are used and that in 50% of the cases these models are used already for over 10 years.

8)     My other comments focus on the presentation.

(a) Line 20. The words “climate neutral” should be changed into “climate-neutral”.

(b) Line 29. such as the emission of greenhouse gases and air pollution

(c) Line 31. policy-making

(d) Line 43. how accessibility

(e) Line 50. and location-based social media data

(f) Line 51. real-time

(g) Line 107. the current use

(h) Line 122. calculating

(i) Lines 139, 184, 199, 209, 238, and 243. at a neighbourhood level

(j) Line 172. six were from

(k) Line 184. Two-thirds 

(l) Line 189. One-thirds

(m) Line 190. For example,

(n) Line 196. And higher

(o) Line 201. In the case of 

(p) Line 203. large variation 

(q) Line 211. are low 

(r) Line 214. could help 

(s) Line 234. decision-making 

(t) Lines 239 and 251. user-friendliness

(u)   (u) Line 260. choose the

This suggestion has been taken by the authors.

We edited all the words and sentences based on the suggestions.

Reviewer 4 Report

The analysis of the urban mobility and traffic flow using GIS models is good for future projections of cities development. Data obtained in this way can lead to an optimum urban management and to increase the wellness of the habitants. 

The questionnaire methods of research is well fitted for an increased number of respondents, covering different areas of interest for the study. Respondents coverage related to European space is well considered.

It is recommended to have a comparative analysis between the present study and others on the same subject to reflect the improvement of the model and positive impact in time.

Also, I recommend the improvement of the study conclusions, as a separate section of the paper, possible to include future trends and recommendations for urban mobility planning based on GIS models.

Figure 4 must be improved for a better understanding.

Author Response

Reviewer

Comments and Suggestions

Reply

4

1)     It is recommended to have a comparative analysis between the present study and others on the same subject to reflect the improvement of the model and positive impact in time.

This is the first paper which provides an overview of the use of GIS models in urban mobility planning (as stated in Introduction and Conclusion). This means that a comparative analysis is not possible. However, we added a reference in the discussion to another up-to-date paper on recently developed GIS models (lines 258–261).

2)     I recommend the improvement of the study conclusions, as a separate section of the paper, possible to include future trends and recommendations for urban mobility planning based on GIS-models.

We now present the conclusion as an individual section. In accordance with the aim of our study, the conclusion focuses on the direction of future GIS model development, not on trends in planning.

3)     Figure 4 must be improved for a better understanding.

As indicated in the text (Results and Discussion), the main purpose of this figure is to show the diversity among cities in how the respondents weighted the five factors, highlighting that modelers should consider the differences in local priorities when they develop GIS models for various cities. Fulfilling this purpose, we prefer not to change the figure.

Round 2

Reviewer 2 Report

The authors addressed my previous comments very well.

Reviewer 3 Report

All my concerns have been well addressed.